# The NUPHAC-EU Framework for Nurses’ Role in Interprofessional Pharmaceutical Care: Cross-Sectional Evaluation in Europe

**DOI:** 10.3390/ijerph18157862

**Published:** 2021-07-25

**Authors:** Elyne De Baetselier, Bart Van Rompaey, Nienke E. Dijkstra, Carolien G. Sino, Kevin Akerman, Luis M. Batalha, Maria I. D. Fernandez, Izabela Filov, Vigdis A. Grøndahl, Jana Heczkova, Ann Karin Helgesen, Sarah Keeley, Petros Kolovos, Gero Langer, Sabina Ličen, Manuel Lillo-Crespo, Alba Malara, Hana Padyšáková, Mirko Prosen, Dorina Pusztai, Bence Raposa, Jorge Riquelme-Galindo, Jana Rottková, Francesco Talarico, Styliani Tziaferi, Tinne Dilles

**Affiliations:** 1Centre for Research and Innovation in Care (CRIC), Nurse and Pharmaceutical Care (NuPhaC), Department of Nursing and Midwifery Science, Faculty of Medicine and Health Sciences, University of Antwerp, 2610 Antwerp, Belgium; Bart.VanRompaey@uantwerpen.be (B.V.R.); Tinne.Dilles@uantwerpen.be (T.D.); 2Research Group Care for the Chronically III, University of Applied Sciences Utrecht, 3584 CH Utrecht, The Netherlands; nienke.dijkstra@hu.nl (N.E.D.); carolien.sino@hu.nl (C.G.S.); 3Department of Nursing, Swansea University, Swansea SA2 8PP, UK; k.l.akerman@swansea.ac.uk; 4Health Sciences Research Unit: Nursing (UICISA: E), Nursing School of Coimbra (ESEnfC), 3046851 Coimbra, Portugal; batalha@esenfc.pt (L.M.B.); isabelf@esenfc.pt (M.I.D.F.); 5Higer Medical School, University “St. Kliment Ohridski”, 7000 Bitola, North Macedonia; belafilov@gmail.com; 6Faculty of Health and Welfare, Østfold University College, 1757 Halden, Norway; vigdis.a.grondahl@hiof.no (V.A.G.); ann.k.helgesen@hiof.no (A.K.H.); 7First Faculty of Medicine, Institute of Nursing Theory and Practice, Charles University, 11000 Prague, Czech Republic; jana.heczkova@lf1.cuni.cz; 8Department of Nursing and Clinical Science, Bournemouth University, Bournemouth BH12 5BB, UK; Skeeley@bournemouth.ac.uk; 9Department of Nursing, University of Peloponnese, 22100 Tripolis, Greece; pkolovos@uop.gr (P.K.); stylianitziaferi216@gmail.com (S.T.); 10Medical Faculty, Institute of Health and Nursing Science, Martin Luther University Halle-Wittenberg, 06108 Halle/Saale, Germany; gero.langer@medizin.uni-halle.de; 11Department of Nursing, Faculty of Health Sciences, University of Primorska, 6310 Izola, Slovenia; Sabina.Licen@fvz.upr.si (S.L.); Mirko.Prosen@fvz.upr.si (M.P.); 12Department of Nursing, Faculty of Health Sciences, University of Alicante, 03690 Alicante, Spain; Manuel.Lillo@ua.es (M.L.-C.); jrgrqlm@gmail.com (J.R.-G.); 13ANASTE-Humanitas Foundation, 00192 Rome, Italy; alba.doc@tiscali.it (A.M.); france.talarico12@gmail.com (F.T.); 14Faculty of Nursing and Professional Health Studies, Slovak Medical University in Bratislava, 83101 Bratislava, Slovakia; hana.padysakova@szu.sk (H.P.); jana.rottkova@szu.sk (J.R.); 15Institute of Nursing Sciences, Basic Health Sciences and Health Visiting, Faculty of Health Sciences, University of Pécs, 7621 Pécs, Hungary; dorina.pusztai@etk.pte.hu (D.P.); bence.raposa@etk.pte.hu (B.R.)

**Keywords:** nursing, medicines management, medicines optimization, patient safety, interprofessional collaboration, nurses’ responsibility, nurses’ tasks

## Abstract

Clear role descriptions promote the quality of interprofessional collaboration. Currently, it is unclear to what extent healthcare professionals consider pharmaceutical care (PC) activities to be nurses’ responsibility in order to obtain best care quality. This study aimed to create and evaluate a framework describing potential nursing tasks in PC and to investigate nurses’ level of responsibility. A framework of PC tasks and contextual factors was developed based on literature review and previous DeMoPhaC project results. Tasks and context were cross-sectionally evaluated using an online survey in 14 European countries. A total of 923 nurses, 240 physicians and 199 pharmacists responded. The majority would consider nurses responsible for tasks within: medication self-management (86–97%), patient education (85–96%), medication safety (83–95%), monitoring adherence (82–97%), care coordination (82–95%), and drug monitoring (78–96%). The most prevalent level of responsibility was ‘with shared responsibility’. Prescription management tasks were considered to be nurses’ responsibility by 48–81% of the professionals. All contextual factors were indicated as being relevant for nurses’ role in PC by at least 74% of the participants. No task nor contextual factor was removed from the framework after evaluation. This framework can be used to enable healthcare professionals to openly discuss allocation of specific (shared) responsibilities and tasks.

## 1. Introduction

Patient safety is an important global health concern. More than twenty years after the publication of the Institute of Medicine’s report *To Err is Human*, serious efforts have been undertaken to decrease the number of medication errors [1,2,3,4,5,6]. In 2017, the World Health Organization’s (WHO) third “Global Patient Safety Challenge on Medication Safety” invited WHO Member States to prioritize medication safety at the national level. The Challenge aimed to make improvements at each stage of the medication process, including prescribing, dispensing, administering, monitoring and use. The target was to reduce severe, avoidable harm resulting from errors or unsafe practices due to weaknesses in health systems by 50% by 2022. The success of this Challenge will depend on the high prioritization of medication safety within healthcare systems globally [7].

Several studies corroborated that pharmaceutical care (PC) can have a serious impact on medication safety and patient-reported outcomes [8,9,10,11]. In the randomized trial of Dürr et al. (2021), the intervention group received an intensified clinical pharmacological/pharmaceutical care, which included medication management and structured patient counseling. Considerable positive effects on the amount of medication errors, patient treatment perception, and severe side effects were shown [6].

One of the opportunities to improve PC and medication safety is strengthening interprofessional collaboration in PC [12,13,14,15,16,17]. Research suggests that an interprofessional team approach, involving pharmacists, physicians and nurses, has the potential to improve team drug-therapy decision-making, continuity of care and patient safety [18]. A review by Donovan et al. (2018) substantiated that a robust body of data supports improvement in patient outcomes when care is provided by an interprofessional team [19]. This interprofessional team approach can enable nurses to raise concerns with physicians and pharmacists, which can contribute to medication error reduction [20,21]. Furthermore, collaboration problems, such as imbalances of authority, professional boundary friction and limited understanding of others’ roles and responsibilities threaten patient safety [22,23]. If role clarity is missing in a team, then effective interprofessional collaboration cannot be guaranteed [24]. After all, poorly defined roles can lead to conflicts in healthcare teams, which negatively effects patient care and patient outcomes [25]. Nowadays, a clear role description of all professionals involved in PC is not always available [21,26,27]. In particular, nurses’ roles are not always explicit, distinct and clear to other professionals, complicating interprofessional collaboration [28,29,30,31]. According to the National Interprofessional Competency Framework of the Canadian Interprofessional Health Collaborative physicians, pharmacists and nurses must understand not only their own roles but also those of other practitioners in the team [32]. The need for a transparent framework describing nurses’ roles in PC is therefore indispensable and urgently needed.

The European Commission funded DeMoPhaC project (DEvelopment of a MOdel for nurses’ role in interprofessional PHArmaceutical Care in Europe) investigates the role of nurses’ in interprofessional PC in 14 European countries. Within this project several large-scale quantitative and qualitative studies are being undertaken with healthcare professionals and nursing students. The overall aim of the project is the development of a framework for nurses’ role in interprofessional PC and the development of an assessment to evaluate nursing curricula and nursing students’ competences in PC. The project started in December 2017. The first part focused on the current clinical practice of nurses. This cross-sectional study showed that monitoring medicines effects, monitoring medicines adherence, nurse prescribing and providing patient education are part of the activities of nurses in clinical practice. Moreover, healthcare professionals felt that nurse involvement should be extended [33]. The second DeMoPhaC study was a qualitative interview study. Healthcare professionals confirmed the positive impact on care quality and patient outcomes when nurses assumed PC responsibilities. The study evidenced the need for a unique and consensus-based PC framework across Europe [34]. In the subsequent scoping review of international literature related to PC by nurses, an overview was given of the variety within nurses’ responsibilities and tasks in PC. Main areas of responsibility were management of therapeutic and adverse effects of medication, medication adherence, patient medication self-management, patient education and information about medication, prescribing, medication safety, and (transition of) care coordination. The extensiveness of nurses’ activities showed nurses to be key persons in PC for patients [35]. Only domains beyond preparation and administration of medication were taken into account. Preparation and administration of medication are basic and generally known activities being performed by nurses even before Florence Nightingale laid the foundation of professional nursing in the 19th century, and hence are not a topic of discussion [36].

Because the scoping review showed nurses can be active in several additional PC domains beyond those initially investigated in the DeMoPhaC project, it is unclear whether healthcare professionals would consider all PC tasks to be nurses’ full responsibility in obtaining best quality of care, or a certain level of supervision by physicians or pharmacists would be required. Additionally, the minimum level of nurse education necessary to perform certain PC tasks has not yet been investigated.

The results of the first three DeMoPhaC studies offer the opportunity to create a framework for nurses’ responsibilities and tasks in PC, together with potential barriers or enablers of nurses performing these PC activities. After the development of such a framework, the content should be evaluated. Therefore, the aim of this study is to create and evaluate a framework describing potential tasks for nurses in PC and to evaluate to what extent physicians, pharmacists and nurses from 14 European countries consider PC-related tasks beyond preparation and administration of medicines to be nurses’ responsibility in an ideal healthcare situation with best quality of interprofessional care and patient outcomes.

## 2. Materials and Methods

### 2.1. Study Design

This observational, descriptive research has a quantitative, cross-sectional study design. The collection of cross-sectional data at a certain point in time allowed us to gather a considerable amount of information from a large pool of participants. The study is reported according to the “Strengthening the Reporting of Observational Studies in Epidemiology” (STROBE) Statement [37] (Appendix A). In an international setting, nurses, physicians and pharmacists were invited to complete an online structured questionnaire on nurses’ tasks within seven pharmaceutical care domains.

### 2.2. Participants and Setting

The study took place in 14 European countries: Belgium, Czech Republic, Germany, Greece, Hungary, Italy, Norway, Portugal, Slovakia, Slovenia, Spain, The Netherlands, the Republic of North Macedonia, and the United Kingdom (Wales and England). The countries were selected in an earlier phase of the overarching DeMoPhaC project of which this study is part.

We included nurses, physicians and pharmacists employed in clinical practice (community care, residential care, hospital care and mental healthcare), education, research, and policy making. Professionals in training and students were excluded.

The estimated sample size to obtain a representative framework of nurses’ role in Europe was calculated with the single population proportion formula [38]. The final sample size was 752, assuming a 50% proportion of risk perception (as this would yield the maximum sample size), a 5% margin of error, and 1.96 as the standard score value for a 95% confidence level.

### 2.3. Framework and Survey Development

In this study, PC is defined as the contribution of “Healthcare professionals” to the care of individuals in order to optimize medicines use and improve health outcomes. This definition is based on a combination of the definition of the Pharmaceutical Care Network Europe (PCNE) and the original definition of Hepler and Strand in 1990 [39,40]. The PCNE definition limits PC to the contribution of pharmacists. Because of the broadly recognized need for interprofessional collaboration in PC, and in line with the original definition of Hepler and Strand, the definition used in this study was extended to all healthcare professionals [20,33,41,42].

The results of the previous quantitative cross-sectional study [33] and the qualitative interview study [34] in European nurses, physicians and pharmacists, followed by the scoping review [35] of the literature, resulted in an overview of seven PC domains, 26 tasks for nurses and 20 contextual factors, which were potential barriers or enablers for nurses taking up tasks in PC (Table 1). Based on this overview, a framework, called the NUPHAC-EU framework (Nurse and Pharmaceutical Care Europe), was created.

To evaluate the content of this framework, an English-language questionnaire was developed by the Belgian researchers in this study (EDB, BVR, TD) and validated (face validity) by the consortium of international experts involved in the DeMoPhaC project. Consequently, the questionnaire was adjusted until consensus was reached (Appendix A).

The survey consisted of three main parts. In the first part eight multiple choice questions defined demographics, employment and education. 

The second part consisted of seven matrices with questions about the level of responsibility for nurses performing tasks within each of the seven PC domains (respectively 15, 17, 16, 14, 22, 16 and 16 tasks, Table 1). Respondents were instructed to envision the ideal situation to obtain the best quality of interprofessional care and patient outcomes. This part of the questionnaire was different for two groups of participants, depending on their ability to distinguish between nurse responsibilities based on nurses’ education level. The first group confirmed being able to make this distinction. They were asked to indicate for each of the four European levels of nurse education (level 5–8) [43] whether each task should be a nursing task and, if so, whether this should be under supervision, with shared responsibility, or fully autonomous. Respondents unable to distinguish between levels of education were asked to indicate the level of responsibility (not allowed, under supervision, with shared responsibility or fully autonomous) for nurses in general. For ‘Prescription management’, two extra questions were presented: (1) the extent to which nurses should be allowed to prescribe medicines in order to obtain best quality of care and patient outcomes, and (2) the necessary restrictions to optimize nurse prescribing in an ideal interprofessional healthcare situation. For the first extra question respondents were asked to consider an ideal situation, which could be different from the current situation. The answering options were: no prescribing, dependent prescribing and independent prescribing. Dependent or supplementary prescribers were defined as “prescribers who’s prescribing is based on clinical management plans, which are put in place for individual patients and relate only to the patient named in the plan. Plans are compiled and signed by both the independent medical prescriber (doctor or dentist), and the supplementary (non-medical) prescriber. They must be agreed by the patient or carer” [44]. Independent prescribers were defined as “practitioners responsible and accountable for the assessment of patients with previously undiagnosed or diagnosed conditions and for decisions about the clinical management required, including prescribing” [45]. For the second extra question, respondents had the possibility of selecting multiple answers from a list of eight predefined restrictions: no restrictions; only a restricted list of medicines; only in a specific context, pathology/specialization; only after specific training; only long-term chronic medicines; only low risk medicines; prescription-only medicines only; only in emergency; and only within an individual patient clinical management plan. Other restrictions could be described in a free text field. 

The third part consisted of 20 questions about contextual factors being barriers or enablers for nurses’ roles in interprofessional PC. Respondents had to indicate the factors of their current healthcare context on a scale from −5 (great barrier), through 0 (no influence on nurses’ responsibilities or tasks), to +5 (great enabler).

The questionnaire was translated into all languages of the participating countries by the specific co-authors. In two countries (Belgium and Italy) the instrument was pilot tested as to its applicability by all three professional groups.

### 2.4. Data Collection

The weblink to the questionnaire was emailed to key stakeholders, professional associations, healthcare facilities and professional networks of the researchers in all countries. Nursing faculties as well as interprofessional colleges (Medicine faculties and Pharmacy faculties) initiated data collection. The weblink was placed on university websites, webpages of professional associations and on social media. Each country received monthly updates about the number of participants.

We aimed to reach a representative sample of nurses, physicians and pharmacists in each country. The length of the questionnaire, however, hindered many potential participants from completing the survey. Moreover, our data collection period (December 2019–August 2020) coincided with the start of the Covid-19 pandemic, resulting in less accessibility to healthcare professionals to take part. Therefore, in March 2020, after two months of data collection, we decided to decrease the number of questions showed to each respondent. Especially the second part of the questionnaire was shown to be too time consuming, when all tasks within all PC domains and all levels of nurse education were considered. Hence, we switched to a shorter survey with all questions of parts 1 and 3, and with only four of the seven matrices presented in part 2. For each participant, the online survey program made a random selection of four PC domains to be shown. This resulted in a significant reduction in the time required to complete all questions, while still allowing each domain to be sufficiently studied.

### 2.5. Data Analysis

Respondents who ended the survey during or immediately after the first part of the questionnaire (demographics, employment, education) were excluded from the data analysis because they did not provide data relevant to the research question. Data were analyzed using IBM SPSS Statistics v27.0 (IBM Corp., Armonk, NY, United States). A two-sided level of significance of 0.05 was used. The main outcome variable was the level of responsibility in PC tasks (not allowed, under supervision, with shared responsibility or fully autonomous) that would be assigned to nurses in an ideal situation with best quality of interprofessional care and patient outcomes, from the perspective of physicians, pharmacists and nurses themselves. Discontinuous data were described using frequency distributions; continuous data were described using a mean value, a minimum and a maximum. To evaluate the statistical significance of the differences between the three professional groups or between the 14 countries, χ^2^ test for nominal variables, and Kruskal–Wallis test for ordinal variables were used. Before Kruskal–Wallis tests were executed, a power analysis using G*power (Universität Düsselfdorf, Düsseldorf, Germany) was performed to determine the minimum number of cases in each country [46]. According to the F-test ANOVA for fixed effects with an a priori medium effect size of 0.25, an alfa of 0.05, and a power of 0.8, at least 28 respondents per group were needed. Consequently, if a country had less than 28 responses, it was not included in the calculation of the *p*-value. 

To clearly visualize as much data as possible, two types of matrices were created. In the first matrix type, each cell shows: (1) the percentage of respondents thinking a certain task could be a nursing task within a certain PC domain and performed by a nurse with a certain level of nurse education, and (2) the color of that cell indicating the mode of the level of responsibility (red for “not allowed”, orange for “to be performed under supervision”, yellow for “to be performed with shared responsibility” or green for “to be performed with full autonomy“). In the second matrix type, the same coloring scheme was used. Each cell shows the percentage of respondents considering a certain task to be a nursing task within a certain country, without distinguishing between the seven PC domains. To achieve this, the PC domains were restructured in two ways: either all seven domains were clustered, or a cluster of six PC domains without “prescription management” was considered.

Data analysis on restructured data resulted in apparently increased sample sizes per country, yet these numbers did not refer to unique respondents, but to clustered data of multiple PC domains per respondent.

To evaluate which tasks had to be either included or excluded from the final framework, we chose a 60% cut-off. In other words, we considered a PC task to be excluded from the framework if indicated as “not allowed for nurses” by at least 40% of the respondents in each country. If a task was evaluated as to be excluded in some, but not all countries, it remained in the framework. After all, the performance of each nursing task in clinical practice will have to be considered in combination with all contextual factors, including country-specific prerequisites.

## 3. Results

### 3.1. The NUPHAC-EU Framework for Nurses’ Role in Interprofessional Pharmaceutical Care in Europe

Taking into account the results of a previous quantitative cross-sectional study [33] and a qualitative interview study [34] in European nurses, physicians and pharmacists, followed by a scoping review of the literature [35], together with the responses in the current cross-sectional evaluation, we developed a framework for nurses’ role in interprofessional pharmaceutical care in Europe (Figure 1). The framework consists of several parts. On top of this, the patient and their network are presented. Together with the patient, the family and the informal caregivers, the interprofessional team, consisting of physicians, nurses, pharmacists and other healthcare professionals, communicates and collaborates in order to obtain the best quality of care and patient outcomes. In the middle of the framework, seven PC domains, beyond medication preparation and administration, and 26 tasks of nurses within these domains, are listed. On the bottom, potential levels of autonomy within the PC domains and tasks are shown, ranging from performing tasks under supervision, through shared responsibility, to full autonomy, and being responsible for a more or less restricted list of medicines. Finally, on the left and right side, twenty contextual factors are defined, being potential barriers or enablers of nurses’ tasks in interprofessional PC.

### 3.2. Research Population to Evaluate the NUPHAC-EU Framework

A total of 1385 respondents participated, of whom 68% were nurses, 17% physicians and 15% pharmacists. The majority (86%) of the respondents were employed in seven of the 14 countries: Slovakia, Belgium, Italy, Slovenia, Czech Republic, Spain and Greece. Mean age was 41 years, and 73% of the population was female. Mean years of work experience in healthcare was 18 years, three quarters of the healthcare professionals were employed in a hospital, and 83% had an active role in clinical practice. More detailed population characteristics are presented in Table 2.

### 3.3. Healthcare Professionals’ Opinions about the Level of Nurse Responsibility for Nurses Performing Tasks in Interprofessional Pharmaceutical Care

In the second part of the data collection, the respondents were presented a random selection of four out of seven PC domains, resulting in smaller samples for domain-specific questions. Questions about domains 1 to 7 were answered by respectively 731, 796, 726, 731, 669, 738, and 711 respondents.

#### 3.3.1. Levels of Nurse Responsibility for European Nurses

Looking at healthcare professionals’ opinions without distinguishing between countries or levels of nurse education, we found that, in an ideal situation, in order to obtain best quality of care and patient outcomes, the majority of the respondents would consider all but four PC tasks to be nurses’ responsibility. These four non-considered tasks were specific for the domain ‘prescription management’: determining type or dosage of medicines, initiating medication, adapting of dose and dose titration and deciding on continuation or cessation of medication. For these tasks, 52.4%, 50.5%, 51.6%, and 51.3% of the respondents, respectively, did not consider them to be nurses’ responsibility. Nevertheless, almost half of the respondents did consider these tasks as possible nursing tasks; hence, all 26 predefined tasks were included into the NUPHAC-EU framework.

Percentages of respondents not considering PC tasks to be nurses’ responsibility ranged from 3.6% to 21.7% for tasks within management of therapeutic and adverse effects, from 3.0% to 18.2% for tasks within management of medicines adherence, from 3.1% to 14.2% for tasks within management of patient medication self-management, from 3.7% to 14.9% for tasks within management of patient education and information, from 18.9% to 52.4% for tasks within prescription management, from 4.6% to 16.5% for tasks within medication safety management, and from 5.2% to 18.0% for tasks within transition of care coordination. 

For the majority of the tasks, “shared responsibility” between nurses and other healthcare professionals was seen as the most appropriate level of responsibility. Detailed percentages of the level of responsibility per task (under supervision, with shared responsibility or fully autonomous) are presented in Figure 2 and Appendix A.

Furthermore, opinions on whether or not nurses should perform PC tasks differed significantly between physicians, pharmacists and nurses for almost all tasks (*p* < 0.001, Appendix B Table A1).

#### 3.3.2. The Ideal Level of Nurse Prescribing

More than one-fifth of the nurses considered “independent nurse prescribing” as the ideal level of nurse prescribing, compared to only 1% of the physicians and 4% of the pharmacists. To obtain best quality of care and patient outcomes, most physicians (55%) and pharmacists (58%) believed that nurses should not prescribe, while the majority of the nurses (51%) thought ‘dependent prescribing’ would be the ideal level of nurse prescribing (*p* < 0.001, Table 3). Healthcare professionals’ opinions also differed between countries, as shown in Figure 3 (*p* < 0.001). The country with the most proponents of “no nurse prescribing” was Slovakia (63%), whereas in the UK (Wales and England), the most “independent nurse” prescribers were considered (41%).

If nurse prescribing—whether or not (in)dependent—were to be considered, several restrictions would be needed in order to optimize prescribing: only after specific training (61%), only a restricted list of medicines (54%), only in a specific context or pathology/specialization (43%), only within an individual patient clinical management plan (36%), only low risk medicines (31%), only long-term chronic medicines (30%), only in emergency (23%), prescription-only medicines only (19%). Still, 7% of the respondents thought there were no restrictions needed. (Table A2, Figure 4)

#### 3.3.3. Differences in Levels of Nurse Responsibility between Countries

Opinions of healthcare professionals about the level of responsibility that nurses should have in an ideal situation differed between countries (*p* < 0.001 for all PC tasks). In countries reaching the minimum sample size for all questions, ranges of percentages of respondents considering PC tasks to be nursing tasks were 31–96% (Belgium), 52–96% (Czech Republic), 63–97% (Greece), 75–99% (Italy), 10–99% (Slovakia), 49–92% (Slovenia), and 59–94% (Spain). The lowest percentages were seen for seven tasks (tasks 16-22) that were specific to one single responsibility: prescription management (Table 4). All percentages of healthcare professionals considering PC tasks to be nursing tasks were increased when ‘prescription management’ was not taken into account (Table A3), indicating lower levels of responsibility were assigned to tasks within prescription management. Tasks within prescription management were considered to be nurses’ tasks by 31–80% (Belgium), 51–82% (Czech Republic), 63–91% (Greece), 75–93% (Italy), 10–66% (Slovakia), 49–85% (Slovenia), and 59–90% (Spain) (Table A4). 

In Greece and Italy, all 22 PC tasks were considered to be nurses’ tasks by at least 60% of the respondents. In the Czech Republic (three tasks), Spain (one task), Belgium (five tasks), Slovenia (six tasks) and Slovakia (seven tasks), more than 40% of the respondents did not consider a part of the tasks to be nurses’ tasks in order to obtain best quality of care and patient outcomes. The latter tasks were all defined as being part of prescription management only (Table 5). Because no one task was indicated as being ‘not allowed for nurses’ by at least 40% of the respondents in each country, no tasks were excluded from the NUPHAC-EU framework after the evaluation.

#### 3.3.4. Levels of Nurse Responsibility for European Nurses of Different Educational Levels

Slightly more than half of the respondents indicated that they were able to make a distinction between nurse responsibilities based on nurses’ educational level (53%), where significantly more nurses (62%) were able to distinguish this item compared to physicians (35%) and pharmacists (28%) (*p* < 0.001).

Within this subsample of healthcare professionals, being able to differentiate between levels of nurse education, most respondents indicated that all PC tasks within all PC domains could be performed by nurses of all educational levels. Between 80% and 100% of the respondents considered that PC tasks could be performed by level 5 nurses. These percentages increased for level 6 nurses (89–100%), level 7 nurses (96–100%) and level 8 nurses (98–100%).

Most tasks were considered to be able to be performed fully autonomously by level 8 nurses, and preferably with shared responsibility by level 5, 6, and 7 nurses. Detailed percentages of the level of autonomy per task, per PC domain and per level of nurse education are presented in Appendix A and Table A5.

### 3.4. Contextual Factors of Nurses’ Role in Current Interprofessional PC

Twenty potential barriers or enablers of nurses’ role in interprofessional PC were presented to the participants. Factors were rated both as barriers and as enablers, median scores ranged from 0 (no influence) to +3 (enabler), and means ranged from −0.2 to +1.9. The highest mean scores were seen for “quality of nurse education”, “level of nurse education”, “interprofessional education”, and “collaborative approach between nurses, physicians and pharmacists” (respectively 1.9, 1.8, 1.6, and 1.5). Figure 5 shows all factors were indicated as barriers or enablers of nurses’ role by at least three quarters of the respondents. Therefore, none of these predefined factors were removed as contextual factors from the NUPHAC-EU framework.

Two potential influencing factors were investigated in more detail: the country and nurses’ educational level.

## 4. Discussion

A framework for nurses’ role aiming for the best quality of interprofessional PC and patient outcomes in an ideal healthcare situation was developed. This NUPHAC-EU framework consists of the patient and their personal and professional network, seven PC domains, and 26 tasks within these domains. These tasks could be performed by nurses with varying levels of autonomy, depending on a range of contextual factors. The majority of the healthcare professionals would consider nurses responsible for tasks within six of the seven domains proposed. Within the domain of prescription management, more respondents were reluctant to allow nurses to take up responsibilities. Overall, physicians, pharmacists and nurses considered a shared responsibility level to be the most appropriate level of autonomy for nurses in PC.

When interpreting the results of this study, it is of major importance to recognize that more than half of the participants were nurses. The comparisons between professional groups showed nurses entrusted with higher levels of responsibility to perform PC tasks. This might have distorted our results in favor of nurses’ more positive opinions regarding their own roles and their opinion about the most appropriate level of autonomy in PC. Despite the higher representation of nurses in this sample, we are convinced of the great value of the NUPHAC-EU framework, which aimed to offer healthcare professionals a discussion tool in a wide range of interprofessional PC situations. The level of nurse responsibility for a certain task in a certain healthcare situation can be different between and within countries, depending on the contextual factors. Because of this, no tasks were removed from the framework, even though they were considered to be irrelevant by the majority of professionals in one or more countries. After all, in other countries with other contexts, the same tasks did meet all prerequisites to be allowed for nurses. This underlines the importance of interpreting the framework as a whole, when openly discussing the allocation of specific (shared) responsibilities and tasks.

Most of the comparisons between the opinions of pharmacists, physicians and nurses showed fewer pharmacists would consider nurses taking up responsibilities in PC. This was also seen in the EUPRON study, where the perceived quality of nurses’ competences in PC was rated the lowest by pharmacists, and hence they were less convinced of the positive impact of nurse involvement on PC [33]. Compared to daily collaborations between physicians and nurses, contacts between pharmacists and nurses in healthcare settings are less frequent or even rare [47,48,49]. This lack of familiarity between pharmacists and nurses might explain the higher percentages of pharmacists considering PC tasks not to be suitable for nurses. After all, it is more difficult to understand another professional’s role, when not working directly together with them. Additionally, the fact that PC was described by pharmacists as a pharmacist-only responsibility for decades may have negatively influenced pharmacists’ opinions in this study, explicitly defining the role of nurses in PC [8,9]. It should be stressed that the development of a model for nurses’ role in PC is in no way an intention to take away responsibilities from other professional groups. In contrast, the NUPHAC-EU model is meant to enable interprofessional collaboration by means of greater role transparancy, which has been demonstrated to positively effect care quality and patient outcomes [50,51,52,53,54,55].

Aiken et al. (2003) showed that educational differences in nurses are related to patient outcomes. Surgical patients experienced lower mortality and failure-to-rescue rates in hospitals with higher proportions of nurses educated at the baccalaureate level (=level 6 of EQF) or higher. They suggested that recruiting and retaining bachelor degree nurses could lead to substantial improvements in quality of care [56]. These results can be extended to the opinions about nurses’ responsibilities in our sample of healthcare professionals. As the level of nurse education increased, more professionals considered PC tasks to be nursing tasks with higher levels of autonomy. Our results, however, cannot be generalized to the opinions of all professional groups, since fewer physicians and pharmacists were able to make a distinction between responsibilities based on nurse educational levels. As already discussed, this might have biased our results.

For tasks within prescription management, more hesitancy regarding nurse involvement was seen. This is not a surprising result, given these nursing tasks were traditionally associated with the medical profession only [57]. However, this situation has been changing in recent decades, with an increasing number of countries legally allowing nurses to prescribe certain medications, either dependently or independently [58]. Despite this relatively recent task shifting between physicians and nurses, studies showed the benefits of nurses taking part in prescription management. Nurse prescribing can improve patient outcomes, such as blood pressure [59,60], cholesterol levels [61], HbA1C levels [60,62], medication adherence [63,64], and patients’ quality of life [65]. Nurse prescribing can also enhance patient safety and satisfaction [62,63], and improve care continuity [63]. Next to better patient outcomes, increased job satisfaction for nurses [64,66] and higher cost-effectiveness of healthcare services because of reduced inappropriate service use [66,67] are also linked to nurse prescribing. We therefore call for a more accepting attitude from healthcare professionals towards nurses prescribing medicines within certain boundaries.

### 4.1. Implications for Clinical Practice, Research, Education, and Policy

Nurses, as key personnel in healthcare delivery, play a critical role in patient care, and more specifically, in PC. To establish appropriate interprofessional relationships, it is necessary to provide a framework that allows the building of trust, co-operation and communication [68]. Our NUPHAC-EU framework will increase the awareness of nurses’ (potential) roles, which will allow pharmacists, nurses and physicians to benefit from teamwork [18]. In further research, expert consensus should be sought regarding necessary PC knowledge, skills, and attitudes for nurses. An overview of nurse competencies based on the NUPHAC-EU framework will enable the development of an assessment to evaluate nurse competences in PC, as guidance for evaluating nurse education, and as a tool for nurse educators. The assessment could also be a tool in the strategy of lifelong learning among nurses in clinical practice.

Currently, the training of healthcare professionals remains largely a single discipline, which may reduce the ability to collaborate interprofessionally [69]. Therefore, more interprofessional education should be organized, as well as rigorous research on interprofessional PC to tackle the remaining barriers. The enablers and barriers presented in the NUPHAC-EU framework can help policy makers and nurse managers to gain insights into the prerequisites for nurses’ role in PC. This can support them in developing workforce planning policies and creating adapted contexts for more barrier-free nurse labor mobility, taking into account feasibility, cost-effectiveness, care quality and patient outcomes. After all, the international mobility of nurses is an increasing phenomenon in the EU, as well as worldwide, and several advantages have been described: a balanced supply and demand for the health workforce; foreign-trained health professionals can fill service gaps and nurse shortages; increased cultural diversity; decreased average age to keep salary levels in check; and sending remittances to the less wealthy home countries [70,71].

### 4.2. Strengths and Limitations of the Study

This study has significant strengths. The NUPHAC-EU framework was developed based on the results of two large-scale quantitative and qualitative studies and a scoping review of the literature, followed by a stakeholders’ evaluation. This resulted in a framework adapted to the needs of clinical practice, with insights into the preferences of the interprofessional team in which nurses collaborate on a daily basis. The framework offers opportunities for discussion in clinical practice, collaboration in research, nurse education and labor mobility of nurses and nursing students. To our knowledge, never before have nurses’ responsibilities in 26 PC-related tasks been distinguished between four EQF levels.

Despite the limited number of participants at the national level in some countries, the overall sample size was satisfactory and provided interesting insights into the extent to which European healthcare professionals consider PC-related tasks to be nurses’ responsibility in an ideal healthcare situation with the best quality of interprofessional care and patient outcomes.

This internet survey had limitations. The inclusion or exclusion of countries and respondents was determined by whether they were included in the overarching Erasmus + project. Additionally, this self-selected sample with an unknown response rate might have led to a distortion of the results due to only the most motivated professionals participating. The enormous workload of healthcare professionals at the time of the COVID-19 pandemic forced many clinicians to neglect activities such as completing scientific surveys. The sample also favored more educated, computer-literate professionals, because of the Internet recruitment. In seven counties, i.e., Germany, Hungary, The Netherlands, the Republic of North Macedonia, Norway, Portugal and the UK, there were low response rates. Therefore, our findings may not be as applicable in these parts of Europe. Finally, as with all self-reports, we cannot discount acquiescence response bias [72]. The views of 1385 professionals are important, yet we have to assume that some might have been biased by socially desirable responding.

## 5. Conclusions

This study aimed to evaluate to what extent physicians, pharmacists and nurses from 14 European countries considered PC-related tasks beyond preparation and administration of medicines to be nurses’ responsibility in an ideal healthcare situation with the best quality of interprofessional care and patient outcomes. The developed NUPHAC-EU framework consisted of the patient and their personal and professional network, seven PC domains, and 26 tasks within these domains, which could be performed by nurses with varying levels of autonomy, depending on a range of contextual factors. The majority of healthcare professionals would consider nurses to be responsible for tasks within six of the seven domains proposed. Within the domain of prescription management, more respondents were reluctant to allow nurses to take up responsibilities. Overall, physicians, pharmacists and nurses considered a shared responsibility level as the most appropriate level of autonomy for nurses in PC.

This framework enables healthcare professionals to openly discuss allocation of specific (shared) responsibilities and tasks.

## Figures and Tables

**Figure 1 ijerph-18-07862-f001:**
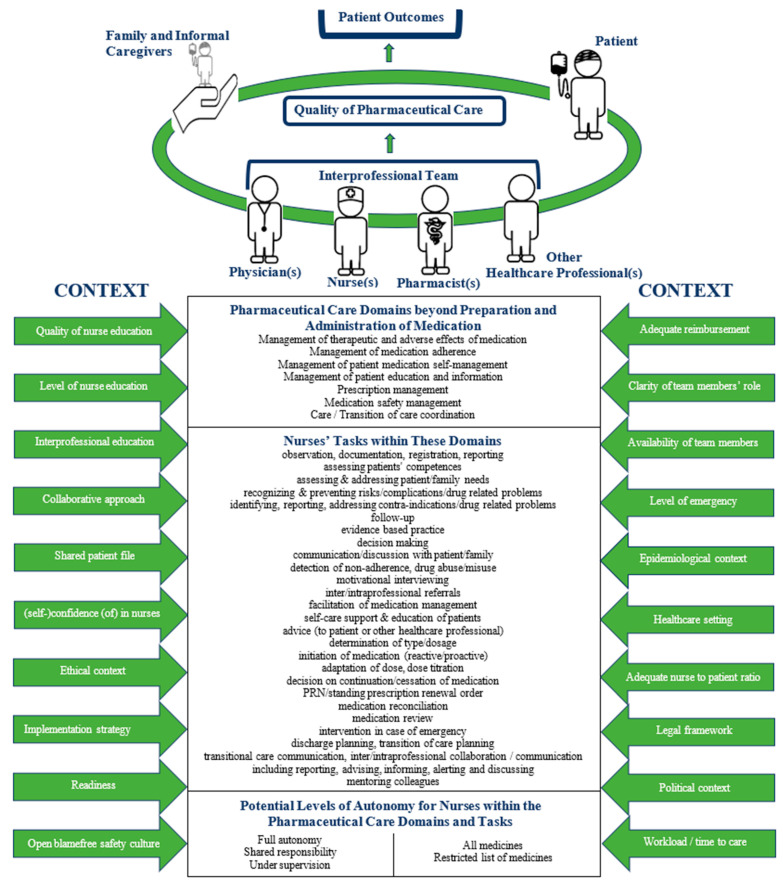
The NUPHAC-EU framework for Nurses’ role in interprofessional Pharmaceutical Care in Europe.

**Figure 2 ijerph-18-07862-f002:**
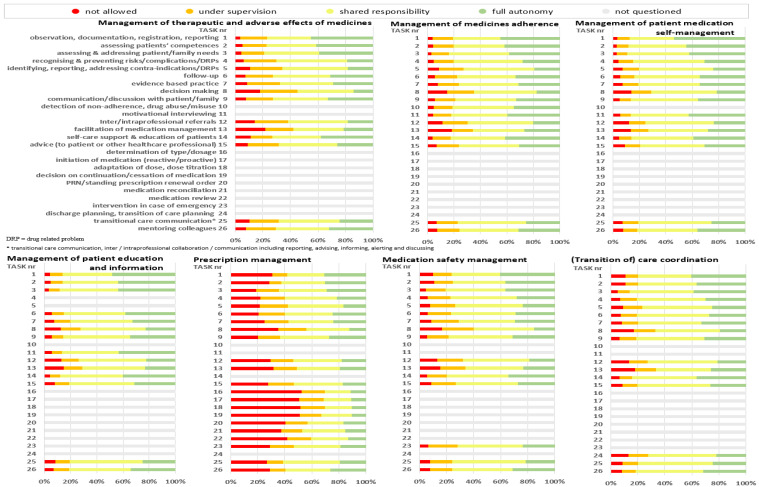
Healthcare professionals’ opinions about the level of nurse responsibility in seven pharmaceutical care domains.

**Figure 3 ijerph-18-07862-f003:**
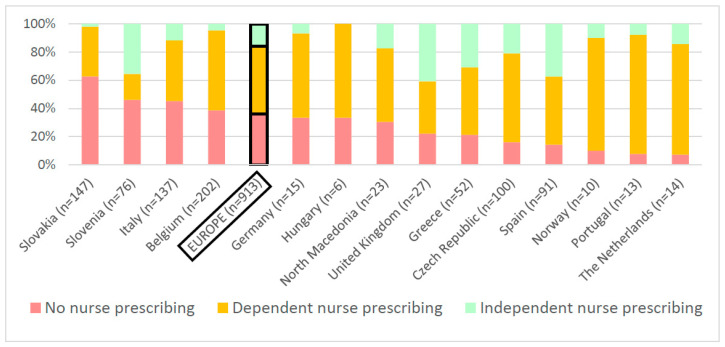
Healthcare providers’ opinion on the level of nurse prescribing authorization in order to obtain best quality of care and patient outcomes in 14 countries (*n* = 913; *p* < 0.001).

**Figure 4 ijerph-18-07862-f004:**
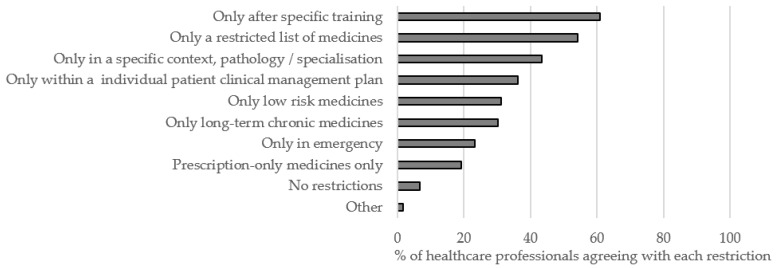
Restrictions to optimize nurse prescribing in an ideal interprofessional healthcare situation (*n* = 537).

**Figure 5 ijerph-18-07862-f005:**
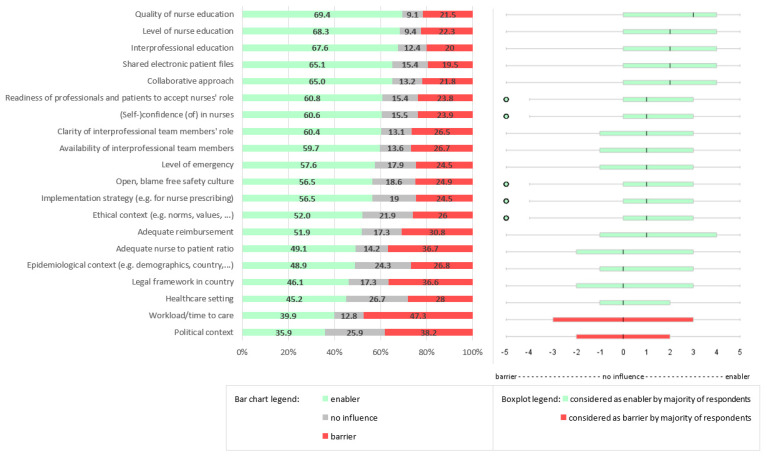
Bar charts (left side) for the percentage of respondents considering 20 contextual factors as barriers or enablers of nurse’ role in interprofessional pharmaceutical care, supplemented with boxplots (right side) for the scores on a 10-point scale from −5 (great barrier) to +5 (great enabler) (*n* = 1005).

**Table 1 ijerph-18-07862-t001:** Overview of 26 potential tasks within 7 pharmaceutical care domains, and 20 contextual factors, for nurses in interprofessional pharmaceutical care, extracted from previous DeMoPhaC studies [33,34,35]. Colors indicate whether the task was part of a pharmaceutical care domain (green) or not (red).

		Domain 1 *	Domain 2 *	Domain 3 *	Domain 4 *	Domain 5 *	Domain 6 *	Domain 7 *
Task 1	Observation, documentation, registration, reporting							
Task 2	Assessing patients’ competences							
Task 3	Assessing & addressing patient/family needs							
Task 4	Recognising & preventing risks/complications/drug related problems							
Task 5	Identifying, reporting, addressing contra-indications/drug related problems							
Task 6	Follow-up							
Task 7	Evidence-based practice							
Task 8	Decision making							
Task 9	Communication/discussion with patient/family							
Task 10	Detection of non-adherence, drug abuse/misuse							
Task 11	Motivational interviewing							
Task 12	Inter/intraprofessional referrals							
Task 13	Facilitation of medication management							
Task 14	Self-care support & education of patients							
Task 15	Advice (to patient or other healthcare professional)							
Task 16	Determination of type/dosage							
Task 17	Initiation of medication (reactive/proactive)							
Task 18	Adaptation of dose, dose titration							
Task 19	Decision on continuation/cessation of medication							
Task 20	PRN/standing prescription renewal order							
Task 21	Medication reconciliation							
Task 22	Medication review							
Task 23	Intervention in case of emergency							
Task 24	Discharge planning, transition of care planning							
Task 25	Transitional care communication, inter/intraprofessional collaboration/communication including reporting, advising, informing, alerting and discussing							
Task 26	Mentoring colleagues							
CONTEXTUAL FACTORS								
Level of emergency		Adequate nurse to patient ratio						
Level of nurse education		Shared digital / electronic patient files and records						
Quality of nurse education		Legal framework in a country						
Interprofessional education		(Self-)confidence in nurses						
Adequate reimbursement		Readiness of healthcare professionals and patients						
Clarity of interprofessional team members’ role		Healthcare setting						
Availability of interprofessional team members		Ethical context (e.g., norms, values, …)						
Workload/time to care		Political context						
Collaborative approach between nurses, pharmacists and physicians		Epidemiological context (e.g., demographics, patient needs, professional needs, …)						

* Domain 1—Management of therapeutic and adverse effects of medicines; Domain 2—Management of medicines adherence; Domain 3—Management of patient medication self-management; Domain 4—Management of patient education and information; Domain 5—Prescription management; Domain 6—Medication safety management; Domain 7—(Transition of) care coordination.

**Table 2 ijerph-18-07862-t002:** Population characteristics (*n* = 1385).

	All (*n* = 1385)	Nurses (*n* = 923)	Physicians (*n* = 240)	Pharmacists (*n* = 199)
Demographical Data	% of Total (*n*)	%	%	%
Country				
Slovakia	18.8 (261)	9.8	35.8	40.7
Belgium	18.2 (252)	15.2	29.2	19.1
Italy	13.4 (186)	15.4	16.3	2
Slovenia	11.0 (153)	13.1	1.7	13.1
Czech Republic	9.3 (129)	11.6	3.3	4.5
Spain	8.4 (117)	10.7	4.6	3
Greece	7.6 (105)	9.8	2.9	3.5
United Kingdom (Wales + England)	3.1 (43)	3.7	1.3	2.5
Republic of North Macedonia	3.0 (41)	3.6	1.7	1.5
Portugal	1.8 (25)	1.7	1.3	3
The Netherlands	1.6 (22)	1.3	0.8	3.5
Germany	1.5 (21)	1.2	1.3	2.5
Norway	1.4 (20)	2.1	-	0.5
Hungary	0.7 (10)	1	-	0.5
Gender				
Female	73.0 (992)	80.2	50	66.8
Male	26.8 (364)	19.4	50	33.2
Other	0.1 (1)	0.1	-	-
Prefer not to say	0.1 (2)	0.2	-	-
Age (years), mean (min-max)	40.8 (18–71)	40.5 (18–71)	42.9 (25–69)	38.7 (23–68)
**Job Characteristics**				
Work experience in HC (years), mean (min-max)	17.5 (0.3–60)	18.0 (0.5–60)	17.1 (0.5–47)	15.7 (0.3–45)
Work experience in HC (setting) ^†^				
Hospital care	74.7 (985)	76.9	81.1	56.7
Community or primary care	26.6 (351)	22.5	20.2	52.9
Residential care	17.1 (225)	21.2	12.6	2.7
Mental healthcare	8.0 (106)	10	3.8	4.3
Current employment ^†^				
Clinical practice	83.2 (1078)	81.7	85.8	86.7
Education	23.5 (304)	26.8	18.8	14.4
Research	12.1 (157)	9.6	22.6	10.8
Policy making	10.5 (136)	10.7	11.7	8.2
**Educational Characteristics**				
Highest level of nursing education (EQF)	Only nurses questioned			
Level 5		24.9		
Level 6		42.1		
Level 7		26.6		
Level 8		6.4		

^†^ More than one answer possible. HC = healthcare. EQF = European Qualifications Framework [43].

**Table 3 ijerph-18-07862-t003:** Physicians’ pharmacists’ and nurses’ opinions about the extent to which nurses should be allowed to prescribe medicines, in order to obtain best quality of care and patient outcomes (=ideal situation, which can be different from the current situation).

Level of Nurse Prescribing	All % (*n*)	Physicians % (*n*)	Pharmacists % (*n*)	Nurses % (*n*)	*p*-Value
No nurse prescribing	36.1 (330)	54.6 (83)	58.1 (68)	27.7 (176)	<0.001
Dependent nurse prescribing	47.9 (437)	44.1 (67)	37.6 (44)	51.0 (324)	
Independent nurse prescribing	16.0 (146)	1.3 (2)	4.3 (5)	21.3 (135)	

*p* Calculated with Kruskal–Wallis test.

**Table 4 ijerph-18-07862-t004:** Percentages of healthcare professionals considering 26 tasks in seven ^$^ pharmaceutical care domains to be nurses’ tasks in order to obtain best quality of care and patient outcomes, split up for 14 countries.

		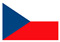	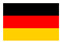	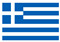	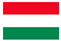		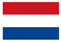	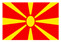		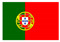	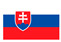	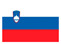	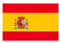	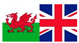	*p*-Value
	Belgium *n* = 622 *	Czech Republic *n* = 199 *	Germany *n* = 56 *	Greece*n* = 193 *	Hungary*n* = 29 *	Italy*n* = 595 *	The Netherlands*n* = 52 *	North Macedonia*n* = 52 *	Norway *n* = 72 *	Portugal*n* = 80 *	Slovakia*n* = 603 *	Slovenia*n* = 350 *	Spain*n* = 295 *	Wales + England *n* = 118 *
T1	94.7	91.5	89.9	96.3	74.2	96.3	97.5	87.3	98.4	98.8	73.4	91.8	92.2	94.7	<0.001
T2	93.8	94.0	86.2	95.9	75.9	96.6	100	86.5	98.4	98.8	76.5	92.0	91.5	92.5	<0.001
T3	93.4	94.1	90.8	95.2	75.9	98.1	96.9	82.0	100	98.8	93.6	90.9	93.7	93.9	<0.001
T4	90.0	91.4	94.6	95.9		97.3	98.0	79.3	100	98.6	88.6	90.3	91.8	96.1	<0.001
T5	82.8	89.2	92.7	95.4		96.2	84.2	70.5	100	98.6	88.7	87.6	89.1	92.2	<0.001
T6	86.3	90.3	92.1	96.9		94.1	96.4	88.1	100	97.5	96.0	88.0	90.3	94.9	<0.001
T7	80.7	88.9	87.5	95.0	75.0	95.1	96.2	77.9	94.5	97.4	92.5	86.5	90.3	94.5	<0.001
T8	66.9	90.9	65.0	86.3		91.4	88.9	60.2	89.1	94.9	81.4	84.5	81.9	88.4	<0.001
T9	89.1	93.4	85.9	91.9	67.9	96.4	97.0	85.7	100	98.8	93.4	89.1	89.6	97.3	<0.001
T10	96.1	90.9		97.9		98.5					96.6	89.2	94.1		<0.001
T11	95.6	94.7	96.6	91.5		97.6	96.8	78.9	100	100	98.6	89.5	91.6	90.7	<0.001
T12	72.6	87.7	58.5	92.3		94.5	89.8	68.4	90.9	98.8	84.9	86.0	85.2	89.8	<0.001
T13	74.8	87.1	87.5	91.7		95.8	96.9	79.3	98.3	98.8	62.5	82.7	82.8	80.0	<0.001
T14	93.1	95.9	98.0	96.6	76.9	97.2	100	93.4	100	100	92.8	88.1	93.2	96.0	<0.001
T15	83.4	89.4	88.7	95.5	75.0	93.5	98.4	79.5	100	98.7	90.7	85.4	87.2	88.6	<0.001
T16	30.7	53.3		72.1		78.0					11.1	51.4	65.6		<0.001
T17	37.7	55.8		68.9		77.1					12.5	51.4	65.1		<0.001
T18	39.7	52.3		66.7		76.6					11.1	52.1	58.7		<0.001
T19	39.4	60.5		62.8		75.4					10.2	50.0	66.7		<0.001
T20	79.0	61.1		76.2		79.8					14.0	49.3	67.7		<0.001
T21	75.4	75.7		76.2		82.0					21.5	53.4	75.0		<0.001
T22	56.6	61.5		67.4		77.3					13.3	76.3	82.1		<0.001
T23	84.4	79.5	71.4	93.0		94.8			77.8	95.2	60.7	86.1	88.9	93.0	<0.001
T24	81.6	80.0		97.6		94.2					88.3	78.8	82.8		<0.001
T25	83.9	90.3	88.7	93.4		95.0	98.0	83.9	98.3	100	86.3	87.3	90.2		<0.001
T26	87.7	88.6	91.4	94.8	75.0	92.6	96.1	80.0	100	98.7	86.9	87.1	87.8	95.1	<0.001

Source country flags: https://countryflags.com, accessed on 3 May 2021. Overview of tasks (T1. T2. … T26): see Table 1 ^$^ seven domains: (1) Management of therapeutic and adverse effects of medicines; (2) Management of medicines adherence; (3) Management of patient medication self-management; (4) Management of patient education and information; (5) Prescription management; (6) Medication safety management; (7) (Transition of) care coordination The colors indicate the level of responsibility that was most prevalent per task (=mode) per country: green = full autonomy; yellow = shared responsibility; orange = under supervision; red = not allowed. Blank cells: no percentage presented because of insufficient valid responses for this task in this country (*n* < 28). *p* calculated with Kruskal–Wallis test for the difference in level of responsibility between countries. Only countries with ≥28 responses were taken into account * *n* = mean number of valid responses. Numbers differ from respondents per country, since tasks were part of several PC domains and are hence shown multiple times.

**Table 5 ijerph-18-07862-t005:** Presentation of 26 tasks within seven ^$^ pharmaceutical care domains considered to be nurses’ tasks in order to obtain best quality of care and patient outcomes by at least 60% of the respondents, split up for 7 countries.

		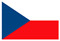	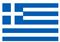		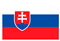	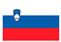	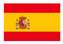	*p*-Value
Belgium*n* = 622 *	Czech Republic*n* = 199 *	Greece *n* = 193 *	Italy*n* = 595 *	Slovakia *n* = 603 *	Slovenia *n* = 350 *	Spain *n* = 295 *
T1	94.7	91.5	96.3	96.3	73.4	91.8	92.2	<0.001
T2	93.8	94.0	95.9	96.6	76.5	92.0	91.5	<0.001
T3	93.4	94.1	95.2	98.1	93.6	90.9	93.7	<0.001
T4	90.0	91.4	95.9	97.3	88.6	90.3	91.8	<0.001
T5	82.8	89.2	95.4	96.2	88.7	87.6	89.1	<0.001
T6	86.3	90.3	96.9	94.1	96.0	88.0	90.3	<0.001
T7	80.7	88.9	95.0	95.1	92.5	86.5	90.3	<0.001
T8	66.9	90.9	86.3	91.4	81.4	84.5	81.9	<0.001
T9	89.1	93.4	91.9	96.4	93.4	89.1	89.6	<0.001
T10	96.1	90.9	97.9	98.5	96.6	89.2	94.1	<0.001
T11	95.6	94.7	91.5	97.6	98.6	89.5	91.6	<0.001
T12	72.6	87.7	92.3	94.5	84.9	86.0	85.2	<0.001
T13	74.8	87.1	91.7	95.8	62.5	82.7	82.8	<0.001
T14	93.1	95.9	96.6	97.2	92.8	88.1	93.2	<0.001
T15	83.4	89.4	95.5	93.5	90.7	85.4	87.2	<0.001
T16	30.7	53.3	72.1	78.0	11.1	51.4	65.6	<0.001
T17	37.7	55.8	68.9	77.1	12.5	51.4	65.1	<0.001
T18	39.7	52.3	66.7	76.6	11.1	52.1	58.7	<0.001
T19	39.4	60.5	62.8	75.4	10.2	50.0	66.7	<0.001
T20	79.0	61.1	76.2	79.8	14.0	49.3	67.7	<0.001
T21	75.4	75.7	76.2	82.0	21.5	53.4	75.0	<0.001
T22	56.6	61.5	67.4	77.3	13.3	76.3	82.1	<0.001
T23	84.4	79.5	93.0	94.8	60.7	86.1	88.9	<0.001
T24	81.6	80.0	97.6	94.2	88.3	78.8	82.8	<0.001
T25	83.9	90.3	93.4	95.0	86.3	87.3	90.2	<0.001
T26	87.7	88.6	94.8	92.6	86.9	87.1	87.8	<0.001

Source country flags: https://countryflags.com (accessed on 3 May 2021). Overview of tasks (T1, T2, … T26): see Table 1. ^$^ seven domains: (1) Management of therapeutic and adverse effects of medicines; (2) Management of medicines adherence; (3) Management of patient medication self-management; (4) Management of patient education and information; (5) Prescription management; (6) Medication safety management; (7) (Transition of) care coordination Green cells indicate the task was considered to be nurses’ task by ≥60% of the respondents (exact % in the cells); red cells indicate the task was not considered to be nurses’ task by >40% of the respondents (exact % in the cells); *p* calculated with Chi^2^ tests for the difference in opinion (whether or not nurses’ task) between countries. * *n* = mean number of valid responses. Numbers differ from respondents per country since tasks were part of several pharmaceutical care domains and are hence shown multiple times. Only countries with ≥28 responses for all 22 tasks were taken into account.

## Data Availability

All data is contained within the article and Appendix A.

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
