# Peer review of "The NUPHAC-EU Framework for Nurses’ Role in Interprofessional Pharmaceutical Care: Cross-Sectional Evaluation in Europe"

_ijerph, 2021, doi:10.3390/ijerph18157862_

Round 1
Reviewer 1 Report
The aim of this study was to create and evaluate a framework describing potential nursing tasks in pharmaceutical care and to investigate nurses’ level of responsibility. A framework of pharmaceutical care tasks and contextual factors was developed. Tasks and context were evaluated using an online survey in 14 European countries. A total of 923 nurses, 240 physicians, and 199 pharmacists responded. The majority considered nurses responsible for tasks within: medication self-management, patient education, medication safety, monitoring adherence, care coordination, and drug monitoring. The most prevalent level of responsibility was ‘with shared responsibility’. Prescription management tasks were considered as nurses’ responsibility by 48%-81% of the professionals. All contextual factors were indicated as relevant for nurses’ role in PC by most participants. No task nor contextual factor was removed from the framework after evaluation. This framework can be used to enable healthcare professionals to discuss allocation of shared responsibilities and tasks.
The study did demonstrate that nurses’ responsibilities could be increased, and this would be acceptable to the healthcare team. This would relieve some burden from physicians for some routine tasks and enhance the job satisfaction of nurses. The study was very thorough and involved 14 countries from Western and Eastern Europe. However, some of the countries had a low level of participation: UK, North Macedonia, Portugal, The Netherlands, Germany, Norway, and Hungary. This would be my only critique of the study and the authors should indicate this as a shortcoming of the study. Obviously, any changes in nursing responsibility would take place at the level of individual countries, so the findings are not as applicable in these countries with a low participation level.
Hopefully, the researchers will follow-up with a study that includes more participants from these 7 countries.
The study is well written, and I found only one typographical error.
Line 69: et al, not et all
Author Response
Point 1: The study was very thorough and involved 14 countries from Western and Eastern Europe. However, some of the countries had a low level of participation: UK, North Macedonia, Portugal, The Netherlands, Germany, Norway, and Hungary. This would be my only critique of the study and the authors should indicate this as a shortcoming of the study.
Response 1: Thank you for the positive feedback. We acknowledge that the findings may not be as applicable in the seven countries with a low response rate. We have added this limitation in the discussion. The following text was added: ‘In seven counties, i.e. Germany, Hungary, The Netherlands, the Republic of North Macedonia, Norway, Portugal and the UK, low response rates existed. Therefore, our findings may not be as applicable in these parts of Europe.’
Point 2: The study is well written, and I found only one typographical error. Line 69: et al, not et all
Response 2: The typographical error in line 69 has been corrected.
Reviewer 2 Report
- At the beginning of the second paragraph, the results of the study by Dürr et all are presented, and it is good for the authors to understand the study briefly what kind of intervention was involved in taking the drug to the patient. Also, please add related references [6].
- It is better to move the PC defined in this study to the Methods section rather than the introduction.
- Lines 100-124 describe the data used in this study. A description of which item was measured in each data should be presented in the Methods section, not in the introduction.
Also, authors need to clearly state when The European Commission funded DeMoPhaC project started. In addition, the previous studies presented through this project should be presented in the introduction section, and the difference between these studies and this study should be clearly shown in the purpose of the study.
- In the study design, the authors only stated that they proceeded according to the STROBE, not the detailed study design. The study design should be presented in detail. In addition, the study method must include the data collection period of participants, exclusion or inclusion criteria, and the sample size required for the study(I think that the sample size should not be presented in statistical methods, but should be presented in the participants.). Also, please indicate the response rate and the number of excluded persons.
The authors said that 28 people are needed for each group in the sample size setting, but there seem to be places where there are fewer than 28 people by country. It would be better to exclude these groups.
- The questionnaire used in the study must be presented with reliability and validity through statistical methods.
- The authors changed the questionnaire items due to time delay during the course of the survey, which must have been measured somewhat differently from the original research plan. I wonder if this survey included the original form or only the changed form. In addition, it is necessary to verify the reliability and validity of the changed questionnaire.
- Authors must present the institution that received the IRB along with ethical considerations.
- In a statistical method, the authors compared responses from each country, and I wonder if the analysis was conducted with weights for participants from each country. If weights were not given, authors would have to re-analyze this into account.
- It is better to present the clinical meaning and applicability of this question developed by the authors in more detail compared to other studies.
- The authors' research has significant results in nursing, but there are some unclear parts in the presentation of the research results, and it seems that they are not presented in a summary. Excessive explanations will be turned to appendix and the overall article will need to be revised.
Round 2
Reviewer 2 Report
I think the revised version has no problems accepting the paper.